# Association of combination statin and antihypertensive therapy with reduced Alzheimer's disease and related dementia risk

Douglas Barthold[1]*, Geoffrey Joyce[2], Roberta Diaz Brinton[3], Whitney Wharton[4,5], Patrick Gavin Kehoe[6], Julie Zissimopoulos[7]

1 Department of Pharmacy, The Comparative Health Outcomes, Policy, and Economics (CHOICE) Institute, University of Washington, Seattle, WA, United States of America, 2 School of Pharmacy, Schaeffer Center for Health Policy and Economics, University of Southern California, Los Angeles, CA, United States of America, 3 Center for Innovation in Brain Science, University of Arizona Health Sciences, Tuscon, AZ, United States of America, 4 School of Nursing, Emory University, Atlanta, GA, United States of America, 5 Department of Neurology, Emory University School of Medicine, Atlanta, GA, United States of America, 6 Bristol Medical School, Translational Health Sciences, University of Bristol, Bristol, United Kingdom, 7 Price School of Public Policy, Schaeffer Center for Health Policy and Economics, University of Southern California, Los Angeles, CA, United States of America

* barthold@uw.edu

**Data Availability Statement:** The primary data source for the project is CMS Medicare claims data. The CMS data used in this project cannot be

## Abstract

### Background

Hyperlipidemia and hypertension are modifiable risk factors for Alzheimer's disease and related dementias (ADRD). Approximately 25% of adults over age 65 use both antihypertensives (AHTs) and statins for these conditions. While a growing body of evidence found statins and AHTs are independently associated with lower ADRD risk, no evidence exists on simultaneous use for different drug class combinations and ADRD risk. Our primary objective was to compare ADRD risk associated with concurrent use of different combinations of statins and antihypertensives.

### Methods

In a retrospective cohort study (2007–2014), we analyzed 694,672 Medicare beneficiaries in the United States (2,017,786 person-years) who concurrently used both statins and AHTs. Using logistic regression adjusting for age, socioeconomic status and comorbidities, we quantified incident ADRD diagnosis associated with concurrent use of different statin molecules (atorvastatin, pravastatin, rosuvastatin, and simvastatin) and AHT drug classes (two renin-angiotensin system (RAS)-acting AHTs, angiotensin converting enzyme inhibitors (ACEIs) or angiotensin-II receptor blockers (ARBs), vs non-RAS-acting AHTs).

### Findings

Pravastatin or rosuvastatin combined with RAS-acting AHTs reduce risk of ADRD relative to any statin combined with non-RAS-acting AHTs: ACEI+pravastatin odds ratio (OR) = 0.942 (CI: 0.899–0.986, p = 0.011), ACEI+rosuvastatin OR = 0.841 (CI: 0.794–0.892, p<0.001), ARB+pravastatin OR = 0.794 (CI: 0.748–0.843, p<0.001), ARB+rosuvastatin OR = 0.818 (CI: 0.765–0.874, p<0.001). ARBs combined with atorvastatin and simvastatin are

shared with other researchers under the terms of our Data Use Agreement (DUA). A researcher can request access to the same data and obtain their own DUA through the CMS Data Request Center (https://www.resdac.org/cms-data/request/cms-data-request-center). The researcher should request Research Identifiable Files. See https://www.resdac.org/cms-data/request/research-identifiable-files. Assistance for accessing and using these data is made available by the Research Data Assistance Center (ResDAC). ResDAC is a consortium of faculty and staff from the University of Minnesota, Boston University, Dartmouth Medical School, and the Morehouse School of Medicine. ResDAC provides free assistance to academic and non-profit researchers interested in using Medicare, Medicaid, SCHIP, and Medicare Current Beneficiary Survey (MCBS) data for research. We will make available the code that is used to generate our analytic data files and conduct the analyses, and anyone will be able to download the code from the repository hosted at https://healthpolicy.box.com/v/Barthold-2019-Statins-AHT-Code. Also included will be a "readme" file that explains how a researcher can get access to the data and a description of the files that will guide a researcher through use of the code.

**Funding:** This research was supported by National Institutes of Health (NIH), awards R01AG055401, P30AG043073, K01AG042498, P01AG026572, R01HL126804, and R01HL130462, and the University of Southern California Zumberge Research Fund, 1R34AG049652. Douglas Barthold was supported through the Schaeffer-Amgen Fellowship Program funded by Amgen. Patrick Kehoe was supported by a Fellowship from the Sigmund Gestetner Foundation, and NIHR-EME, Alzheimer's Society, Alzheimer's Research UK, BRACE, the Medical Research Council, and the British Heart Foundation. Sponsors' role: The sponsors had no role in the design, methods, analysis, interpretation, or preparation of the paper.

**Competing interests:** Partial financial support for this research was provided by Amgen. This does not alter our adherence to PLOS ONE policies on sharing data and materials. The company had no role in any stage of the research process. The concept, design, acquisition, analysis, and interpretation of data, manuscript drafting, and revisions, were completed without any involvement from the company.

associated with smaller reductions in risk, and ACEI with no risk reduction, compared to when combined with pravastatin or rosuvastatin. Among Hispanics, no combination of statins and RAS-acting AHTs reduces risk relative to combinations of statins and non-RAS-acting AHTs. Among blacks using ACEI+rosuvastatin, ADRD odds were 33% lower compared to blacks using other statins combined with non-RAS-acting AHTs (OR = 0.672 (CI: 0.548–0.825, p<0.001)).

## Conclusion

Among older Americans, use of pravastatin and rosuvastatin to treat hyperlipidemia is less common than use of simvastatin and atorvastatin, however, in combination with RAS-acting AHTs, particularly ARBs, they may be more effective at reducing risk of ADRD. The number of Americans with ADRD may be reduced with drug treatments for vascular health that also confer effects on ADRD.

## Introduction

Alzheimer's disease and related dementias (ADRD) are a large and growing public health issue. There are approximately 7 million individuals aged 65 and older living with ADRD in the United States, and this number is projected to grow to 12 million by 2040 [1]. With no disease-modifying treatments for ADRD, there has been increased global attention on prevention and risk reduction of ADRD by maintenance of a healthy lifestyle and management of diseases such as hypertension and dyslipidemia [2–5]. A reduction in risk of ADRD would bring significant health and financial benefits to individuals, societies, and families [6].

Antihypertensive therapies are effective pharmaceutical treatments for hypertension, and statins for dyslipidemia. Statins are hypothesized to act on Alzheimer's disease (AD) pathology through their effect on cholesterol as well as through other nonlipid effects such as reduced inflammation, although heterogeneity in molecular structure, efficacy, and pharmacokinetics across statins suggests potential variability in effects [7–12]. Randomized controlled trials and reviews in this area have failed to find a beneficial effect of statins on AD, although limitations of these studies (e.g., short follow-up periods, non-representative samples, and removal of hyperlipidemic subjects) are well recognized [13–15]. There is, however, a growing body of observational evidence showing that statins are associated with reduced ADRD risk [16–22], including our own work that featured robust control for confounding in a large and representative dataset [16]. Recent evidence also suggests that some subgroups may be more responsive to statin therapy than others [23, 24]. Though potential influence of statins on AD pathology is still unclear, some trials suggest statin-related attenuation of tau in middle age participants, which aligns with the findings from a community based-autopsy cohort [25, 26].

Systematic analysis of observational, randomized controlled trials, and meta-analyses found AHT treatment was protective against cognitive decline and incident dementia [27–30]. Several studies have implicated the renin-angiotensin system (RAS) in AD, in particular over-activation of the *classical* hypertension-promoting RAS (cRAS), and more recently dysfunction of the *regulatory* RAS (rRAS), which ordinarily regulates cRAS activity [31–37]. A recent study confirmed these results and reported that RAS AHT users (vs. other AHT users) exhibited fewer neurofibrillary tangles in predetermined brain regions involved in AD [38]. These findings are consistent with hypotheses that therapeutic classes of AHTs operating to reduce RAS activity (i.e., angiotensin converting enzyme inhibitors (ACEIs) and angiotensin-II receptor blockers (ARBs)), may affect AD differently than other AHT drug classes (e.g., calcium

channel blockers, beta blockers, loop diuretics, or thiazide diuretics) [39–41]. Moreover, in vitro and animal studies of AD show that ARBs were more protective than ACEIs because of ACEI's potential role in amyloid-beta degradation, and in Tg2576 mice that enhancement of rRAS can prevent and reverse cognitive decline and reduce amyloid-beta pathology [35, 40, 42, 43]. Our recent observational study confirmed this relationship in a longitudinal, population study [44].

Approximately one quarter of older Americans used both a statin and an AHT in 2014. Despite the high prevalence of use and hypothesized variation in effects across different types of statins and AHTs, there are no existing analyses of the association of use of specific combinations of AHT and statins and ADRD risk. The widespread use of these drugs increases the importance of analyses examining the combinations of therapies that are associated with the greatest and least reduction in ADRD risk and for specific populations. The purpose of this study is to use data from a large population-based cohort to examine the hypothesis that some combinations of statins and AHTs may be more protective against ADRD onset than others, and analyze variation by sex and race/ethnicity in order to inform about the potential for population-specific health benefits.

## Methods

### Data

We examined the medical and pharmacy claims of a random 20% sample of Medicare beneficiaries, aged 67 and higher, enrolled in traditional Medicare (fee-for-service) from 2007 to 2014. Medicare is publicly funded health insurance in the United States for individuals aged 65 and higher, as well younger beneficiaries with certain permanent disabilities. As of 2018, there were approximately 60 million enrollees in Medicare, two thirds of whom had traditional (fee-for-service) Medicare (as opposed to Medicare Advantage, a privately administered alternative) [45]. Traditional Medicare tends to enroll slightly less healthy beneficiaries than Medicare Advantage, but still comprise a majority of older adults in the United States, with nationwide representativeness [45]. We linked claims from Medicare Parts A (inpatient care), B (outpatient care), and D (prescription drugs) to enrollment files that included beneficiaries' characteristics. Part D claims include key elements related to prescription drug utilization, while Parts A and B claims capture include detailed diagnosis and procedure codes (*International Classification of Diseases*, *Ninth Revision* (ICD-9)). These data were further supplemented with claims histories from the Chronic Conditions Warehouse. Institutional review board approval was granted by the University of Southern California University Park IRB, which granted a waiver of participant consent under 45 CFR 46.116(d).

### Study sample

The study sample consisted of person-years for Medicare beneficiaries age 67 and older. In each person-year, we required observation for a minimum of three years (that year and the previous two) with consecutive fee-for-service enrollment, Part D enrollment, and no death. Individuals were required to have used both an AHT and a statin for the two previous years, have no prior ADRD diagnoses, and no prior use of AD specific medications (acetylcholinesterase inhibitors (AChEIs) or memantine). ICD-9 codes are specified in S1 Appendix. The analytic sample consisted of 694,672 unique beneficiaries, followed for a total of 2,017,786 person-years (1,241,491 for females, and 776,295 for males). Table 1 shows the characteristics of the analytic sample.

**Table 1. Characteristics of combination statin and AHT users in 2009–2014 Medicare claims data.**

| | RAS + any statin | Non-RAS AHT + any statin | ACEI + ator | ACEI + pra | ACEI + rosu | ACEI + sim | ARB + ator | ARB + pra | ARB + rosu | ARB + sim |
|---|---|---|---|---|---|---|---|---|---|---|
| ADRD | 2.07% | 2.64% | 2.18% | 2.03% | 1.63% | 2.16% | 2.02% | 1.94% | 1.62% | 2.02% |
| Age | 77.3 | 78.0 | 77.2 | 77.2 | 76.2 | 77.1 | 77.6 | 77.8 | 76.7 | 77.6 |
| Female | 61% | 63% | 56% | 60% | 56% | 57% | 66% | 70% | 67% | 68% |
| White | 83% | 86% | 84% | 86% | 84% | 85% | 78% | 82% | 78% | 79% |
| Black | 6% | 6% | 6% | 7% | 7% | 6% | 7% | 7% | 6% | 7% |
| Hispanic | 6% | 4% | 6% | 4% | 6% | 5% | 7% | 5% | 8% | 7% |
| Other race | 5% | 4% | 4% | 3% | 3% | 3% | 9% | 6% | 8% | 7% |
| % HS grad | 76% | 76% | 76% | 75% | 75% | 76% | 76% | 76% | 75% | 76% |
| Median income | $55,745 | $55,919 | $56,728 | $53,436 | $54,187 | $54,208 | $59,299 | $55,548 | $56,784 | $56,319 |
| # physician visits | 9.5 | 9.9 | 9.1 | 9.2 | 9.9 | 8.7 | 10.7 | 10.8 | 11.6 | 10.1 |
| HCC | 1.25 | 1.33 | 1.26 | 1.24 | 1.25 | 1.24 | 1.28 | 1.26 | 1.26 | 1.26 |
| AMI | 9% | 9% | 10% | 8% | 10% | 9% | 8% | 7% | 8% | 7% |
| ATF | 18% | 23% | 18% | 19% | 18% | 17% | 18% | 19% | 17% | 17% |
| Diabetes | 53% | 40% | 52% | 51% | 54% | 53% | 55% | 52% | 57% | 54% |
| Stroke | 16% | 18% | 16% | 17% | 16% | 16% | 17% | 17% | 17% | 16% |
| Beneficiaries | 507,304 | 279,398 | 92,551 | 48,545 | 35,128 | 173,653 | 60,569 | 29,250 | 27,259 | 95,007 |
| Observations | 1,387,009 | 630,777 | 214,734 | 106,203 | 78,620 | 435,902 | 140,609 | 62,749 | 61,080 | 231,174 |

Sample of 2009–2014 Medicare person-years with 90 possession days and 2 claims of both an AHT and a statin in both years t-1 and t-2. RAS (renin-angiotensin system) AHTs are antihypertensive (AHT) prescription drugs (angiotensin converting enzyme inhibitors (ACEIs) and angiotensin-II receptor blockers (ARBs). Non-RAS acting AHTs are beta-blockers, calcium channel blockers, loop diuretics, and thiazide diuretics. Statins are atorvastatin, pravastatin, rosuvastatin, and simvastatin. Sample restricted to person-years with 3 years fee-for-service, 3 years Part D, age 67+, no deaths in the reference year (year t), no prior ADRD diagnoses, and no prior use of acetylcholinesterase inhibitors (AChEIs) or memantine. Abbreviations: ADRD (Alzheimer's disease and related dementias), HS (high school), HCC (Hierarchical Condition Category index), AMI (acute myocardial infarction), ATF (atrial fibrillation).

## Measuring statin and AHT exposure

We identified AHT and statin use in Medicare Part D prescription drug claims (2007–2013). AHT use was for the following classes: angiotensin-converting enzyme inhibitors (ACEI), angiotensin-II receptor blockers (ARB), beta-blockers (BBL), calcium channel blockers (CCB), loop diuretics (LDs), and thiazide-like diuretics (TDs). Statin use was for the following molecules: atorvastatin, pravastatin, rosuvastatin, and simvastatin, which were the four most commonly used molecules in these data. We defined combination AHT and statin users as any individual with 90 days supply and at least two claims in a year for both types of drugs for the two previous years. This threshold was chosen as the minimum necessary to ensure regular use of the drug during the exposure period, and is consistent with the definitions used in earlier evidence [44]. We examined robustness of results to other definitions of use: 180 days and 2 claims, 270 days and 2 claims.

## Study design

In a retrospective cohort design, we compared onset of ADRD separately for RAS-acting ACEI and ARB therapies combined with each different molecule of statin, to onset among persons using other combinations of non-RAS acting AHTs and statins. Onset of ADRD was measured using index date of ADRD diagnosis and we required an ADRD index diagnosis to be verified with either a second ADRD diagnosis code or 'dementia specified elsewhere' diagnosis code in a subsequent claim within the study period (2009–2014). Measurement error in the timing of

ADRD diagnosis is unlikely to bias our results because the same error exists for subjects in both the exposure and comparator groups. To mitigate concern that imminent ADRD onset could lead to poor adherence or discontinuation of AHT use, we designated years *t-1* and *t-2* as the drug exposure years, prior to assessing ADRD risk in year *t*. For example, we relate AHT/statin use in 2007–2008 to ADRD risk in 2009; in the absence of an ADRD diagnosis, the same individual remains in the sample and their AHT/statin use in 2008–2009 is related to their ADRD incidence in 2010, and further, as long as they continue to meet the inclusion/ exclusion criteria. Sex and race/ethnicity specific analyses use only members of the same sex or race/ethnicity as the comparison group; thus estimated differences in association across combinations is attributed to the drugs and not due to unobserved differences across sex or race/ ethnicity.

### Statistical analyses

We examined the association of combination AHT and statin use (in years *t-1* and *t-2*) and incident ADRD (year *t*), during the years 2009 to 2014. We used multivariable logistic regression to control for the potentially confounding roles of age, age squared, sex, race, proxy measures of socio-economic status (high school graduation rate within the beneficiary's zip code (quartiles), zip code median income (quartiles)), years since hypertension and hyperlipidemia diagnoses, comorbidity index (quartiles), number of physician visits (quartiles), and indicators for past diagnoses of diabetes, atrial fibrillation, acute myocardial infarction, and stroke. Time-varying covariates were measured in year *t-1*. Health status was measured with past diagnosis of key comorbidities, as well as the Centers for Medicaid and Medicare Services-Hierarchical Condition Category (CMS-HCC), an index based on health status from diagnostic data and demographics, in which higher numbers indicate worse health. The CMS uses this index to predict health expenditures in the next year, and it correlates highly with mortality [46]. We used years since hypertension and hyperlipemia diagnoses, as measured by the Chronic Conditions Warehouse (CCW) as far back as 1999, to control for unobserved AHT and statin use in the years prior to Medicare Part D enactment in 2006. Race/ethnicity was determined with the beneficiary race code in CMS enrollment data, and with the application of a name-based identification algorithm from the Research Triangle Institute [47]. Standard errors were clustered at the county level. We ran analyses for the sample as a whole, as well as for each sex and for different racial/ethnic populations.

### Results

Table 1 depicts the characteristics of our analytic sample. The annual ADRD incidence rate in this sample is 2.07% among persons using a RAS-acting AHT and any statin, and 2.64% among persons using a non-RAS-acting AHT and any statin. Generally, rates were lower persons using ARBs compared to ACEI with the exception of no difference in rates among persons using them in combination with rosuvastatin. Those using a RAS-acting AHT were younger, more likely to be male, non-white, and have a lower mean HCC score. There were no differences in percent high school graduate or median income. ACEI users compared to ARB users, across statin types, are about 0.5 years younger, more likely to be male, white, and have fewer physician visits, and lower HCC scores. There were no differences in percent high school graduate or median income.

Fig 1 reports odd ratios (ORs) and 95% confidence intervals from logistic regressions of ADRD incidence that adjust for the patient and setting characteristics described above. Compared to individuals using statins combined with non-RAS AHTs, use of pravastatin and rosu-vastatin in combination of ARB or ACEI was associated with reduced risk of ADRD (ACEI

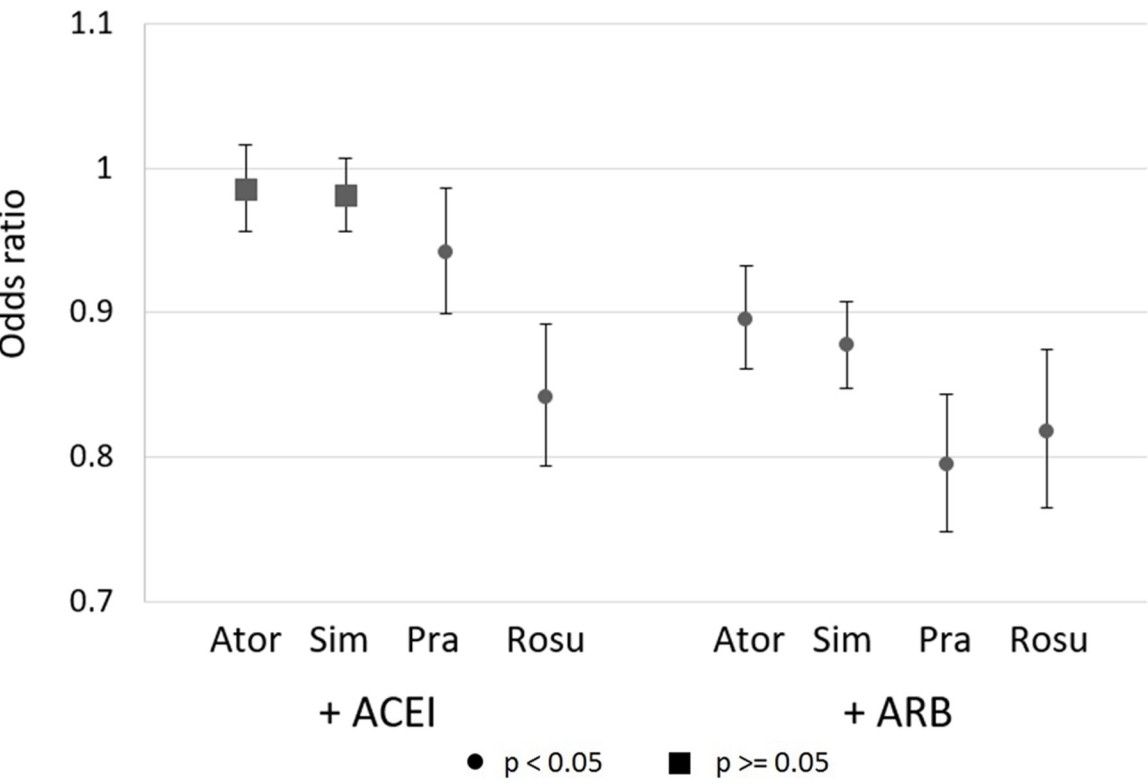

**Fig 1. Adjusted odds ratios of ADRD incidence associated with use of statin-AHT combinations, relative to users of other statin-AHT combinations, with 95% confidence intervals.** Logistic regression results for ADRD incidence in sample of 2009–2014 Medicare person-years (N = 2,017,786) with 90 possession days and 2 claims of both an AHT and a statin in both years t-1 and t-2. AHTs are antihypertensive (AHT) prescription drugs (angiotensin converting enzyme inhibitors (ACEIs), angiotensin-II receptor blockers (ARBs), beta-blockers, calcium channel blockers, loop diuretics, and thiazide diuretics), and statins are atorvastatin, pravastatin, rosuvastatin, and simvastatin. Sample restricted to person-years with 3 years fee-for-service, 3 years Part D, age 67+, no deaths in the reference year (year t), no prior ADRD diagnoses, and no prior use of acetylcholinesterase inhibitors (AChEIs) or memantine. Controls are age, age squared, sex, education, income quartiles, statin use (t-1), years since hypertension and hyperlipidemic diagnoses, HCC comorbidity index, number of physician visits, and indicators for past diagnoses of diabetes, atrial fibrillation, acute myocardial infarction, and stroke. Standard errors are clustered at the county level.

+pravastatin OR = 0.942 (CI: 0.899–0.986, p = 0.011), ACEI+rosuvastatin OR = 0.841 (CI: 0.794–0.892, p<0.001), ARB+pravastatin OR = 0.794 (CI: 0.748–0.843, p<0.001), ARB+rosuvastatin OR = 0.818 (CI: 0.765–0.874, p<0.001)). There was no reduced risk associated with ACEI use and either atorvastatin or simvastatin, but a reduced risk when combined with ARBs (ARB+atorvastatin OR = 0.896 (CI: 0.861–0.932, p<0.001), ARB+simvastatin OR = 0.877 (CI: 0.848–0.908, p<0.001)).

Table 2 reports results of ORs separately for men and women. Combined use of ACEIs with three statins (atorvastatin, simvastatin, and pravastatin), compared to combined use statins and non-RAS-acting AHTs, was not associated with reduced ADRD risk among women. ACEI+rosuvastatin and ARB+rosuvastatin were associated with lower odds of ADRD among women (ACEI+rosuvastatin OR = 0.832 (CI: 0.776–0.892, p<0.001), ARB+rosuvastatin OR = 0.833 (CI: 0.774–0.895, p<0.001). Among men, with the exception of simvastatin, combine use of any statin and either ACEI or ARB was associated with lower odds of ADRD incidence.

Table 2 also reports results of ORs separately by race/ethnicity. Among Hispanics, there is no combination of statins and RAS-acting AHTs compared to combinations of statins and

**Table 2. Adjusted odds ratios of ADRD incidence associated with use of statin-AHT combinations, relative to users of other statin-AHT combinations.**

| Statin | AHT | | Female | Male | White | Black | Hispanic |
|--------|-----|-----|--------|------|-------|-------|----------|
| Ator | ACEI | OR | 1.021 | 0.916 | 0.984 | 0.983 | 1.009 |
| | | CI | (0.983–1.060) | (0.868–0.966) | (0.951–1.019) | (0.864–1.117) | (0.899–1.132) |
| | | p | 0.291 | 0.001 | 0.365 | 0.788 | 0.878 |
| Sim | ACEI | OR | 0.989 | 0.962 | 0.981 | 0.946 | 0.941 |
| | | CI | (0.957–1.022) | (0.921–1.005) | (0.953–1.009) | (0.872–1.027) | (0.862–1.028) |
| | | p | 0.499 | 0.082 | 0.180 | 0.187 | 0.180 |
| Pra | ACEI | OR | 0.981 | 0.854 | 0.944 | 0.877 | 0.958 |
| | | CI | (0.927–1.038) | (0.787–0.926) | (0.897–0.992) | (0.738–1.043) | (0.784–1.169) |
| | | p | 0.504 | <0.001 | 0.024 | 0.137 | 0.671 |
| Rosu | ACEI | OR | 0.832 | 0.858 | 0.848 | 0.672 | 0.902 |
| | | CI | (0.776–0.892) | (0.777–0.949) | (0.795–0.905) | (0.548–0.825) | (0.753–1.080) |
| | | p | <0.001 | 0.003 | <0.001 | <0.001 | 0.261 |
| Ator | ARB | OR | 0.902 | 0.883 | 0.902 | 0.869 | 0.869 |
| | | CI | (0.862–0.945) | (0.819–0.953) | (0.863–0.942) | (0.757–0.999) | (0.744–1.014) |
| | | p | <0.001 | 0.001 | <0.001 | 0.048 | 0.074 |
| Sim | ARB | OR | 0.893 | 0.837 | 0.878 | 0.856 | 0.898 |
| | | CI | (0.860–0.928) | (0.785–0.893) | (0.846–0.911) | (0.763–0.960) | (0.802–1.005) |
| | | p | <0.001 | <0.001 | <0.001 | 0.008 | 0.060 |
| Pra | ARB | OR | 0.824 | 0.703 | 0.797 | 0.770 | 0.877 |
| | | CI | (0.771–0.881) | (0.615–0.804) | (0.746–0.852) | (0.620–0.956) | (0.685–1.121) |
| | | p | <0.001 | <0.001 | <0.001 | 0.018 | 0.294 |
| Rosu | ARB | OR | 0.833 | 0.784 | 0.798 | 0.902 | 0.842 |
| | | CI | (0.774–0.895) | (0.684–0.900) | (0.742–0.858) | (0.735–1.106) | (0.692–1.025) |
| | | p | <0.001 | 0.001 | <0.001 | 0.320 | 0.086 |
| N | | | 1,241,491 | 776,295 | 1,689,066 | 125,843 | 106,019 |

Logistic regression results for ADRD incidence in sample of 2009–2014 Medicare person-years with 90 possession days and 2 claims of both an AHT and a statin in both years t-1 and t-2. AHTs are antihypertensive (AHT) prescription drugs (angiotensin converting enzyme inhibitors (ACEIs), angiotensin-II receptor blockers (ARBs), beta-blockers, calcium channel blockers, loop diuretics, and thiazide diuretics), and statins are atorvastatin, pravastatin, rosuvastatin, and simvastatin. Sample restricted to person-years with 3 years fee-for-service, 3 years Part D, age 67+, no deaths in the reference year (year t), no prior ADRD diagnoses, and no prior use of acetylcholinesterase inhibitors (AChEIs) or memantine. Controls are age, age squared, sex, education, income quartiles, statin use (t-1), years since hypertension and hyperlipidemic diagnoses, HCC comorbidity index, number of physician visits, and indicators for past diagnoses of diabetes, atrial fibrillation, acute myocardial infarction, and stroke. Standard errors are clustered at the county level.

non-RAS acting AHTs that is associated with lower risk of ADRD. Among blacks, the reduced risk associated with combined use of rosuvastatin and ACEI is large (OR = 0.672, CI: 0.548–0.825, p<0.001). Among blacks there is also reduced risk associated with combined use of simvastatin and ARBs (OR = 0.856, CI: 0.763–0.960, p = 0.008), atorvastatin and ARBs (OR = 0.869, CI: 0.757–0.999, p = 0.048), and pravastatin and ARBs (OR = 0.770, CI: 0.620–0.956, p = 0.018). Among whites, combined use of ARBs and any statin is associated with lower odds of ADRD relative to non-RAS acting AHTs and any statin.

## Discussion

In this study, we examined the association between use of combined statin and RAS-acting AHT drug therapies and ADRD risk, compared to risk associated with combined statin and non-RAS acting AHT therapy. RAS-acting AHTs were disaggregated into ACEIs and ARBs, and statins were disaggregated by molecule (atorvastatin, pravastatin, rosuvastatin, and

simvastatin). No previous study has investigated the association of different combinations of these frequently used drugs and ADRD risk in a large and broadly representative sample of older Americans and across sex and racial and ethnic subpopulations. Among elderly adults, who were using drugs to treat hypertension and dyslipidemia we found substantial variation in ADRD risk reductions across different combinations of statins and AHTs, and across different populations. The greatest reductions, as compared to individuals using statins and non-RAS acting AHTs, were for individuals using ARBs in combination with pravastatin or rosuvastatin. These results were robust to varying definitions of drug use (90, 180, and 270 days) (S2 and S3 Tables). This finding held in some subpopulations: women, men, whites, and blacks. Our detection of significant reduction in ADRD risk for black men using specific combinations of statins and RAS-acting AHTs is important, as this group was not previously found to benefit from either statin therapy or RAS-acting AHT therapy [16, 44].

The magnitudes of estimated risk reductions were meaningful; for example, using ARBs combined with pravastatin was associated with 21% lower odds of ADRD, as compared to individuals using other statins and non-RAS acting AHTs in combination (OR = 0.794, CI: 0.748–0.843, p<0.001). As for users of ACEIs, the other RAS-acting AHT, only in combination with rosuvastatin was there an associated reduction in risk of ADRD among older American populations of women, men, whites, and blacks as compared to those same populations using statins and non-RAS acting AHTs. This association was robust to varying definitions of use (90, 180, and 270 days) (S2 and S3 Tables). While our results show greater protection for combinations that involve rosuvastatin or pravastatin, these drugs are not as frequently used as atorvastatin and simvastatin. In 2007, 2.9% of Medicare beneficiaries with fee-for-service (FFS) and Part D used pravastatin, and 3.3% used rosuvastatin, compared to 15.6% for atorvastatin, and 18.1% for simvastatin (Fig 2). Use of rosuvastatin and pravastatin has grown over time to 8.9% and 5.8% in 2014 for pravastatin and rosuvastatin, respectively (Fig 2). Rosuvastatin has shown strong growth in market share since its branded version, Crestor, lost patent protection in the United States in 2016 [48]. In 2017, rosuvastatin was the sixth fastest growing mail and retail prescription in the U.S., with an additional 6.8 million 90-day equivalent prescriptions [49]. The pharmacokinetics and pharmacodynamics of rosuvastatin are distinct from other statins, and may be a contributing factor to the differences we found [50, 51]. Rosuvastatin appears to have preferential benefit over other statins with respect to favorable low-density lipoprotein (LDL) cholesterol-lowering efficacy, as well as lower likelihood of side effects due t low extra-hepatic penetration and interference with cytochrome P450 related drug interactions [51]. In the ARIES trial, rosuvastatin improved the overall lipid profile of hypercholesterolemic African-Americans better than the milligram-equivalent doses of atorvastatin [52]. As the use of these molecules increases, additional benefits may be conferred for older adults of diverse backgrounds.

In this study we cannot test what mechanisms are driving the differential associations that were observed between the four statins investigated, all of which work to reduce low-density lipoprotein (LDL) cholesterol levels, and combinations of ARBs or ACEIs. A potential mechanism, and one that may explain some of the ethnic differences observed, are differences in drug metabolism and transport. Rosuvastatin, for example, is metabolized by CYP2C9 and CYP2C19 cytochrome P-450 enzymes. These CYP enzymes could be related to differential associations with ADRD in two ways. First, if there are pharmacokinetic interactions between certain statins and certain AHTs, the metabolism of one or both drugs could be altered in a way that confers differential benefit regarding ADRD. CYP2C9, for example, metabolizes Rosuvastatin, and is a substrate for several ARBs (e.g., losartan and irbesartan), potentially leading to interactions [53]. Second, statins and/or AHTs may affect the activity of CYP enzymes that have a direct impact on ADRD risk factors. For example, some genetic variants

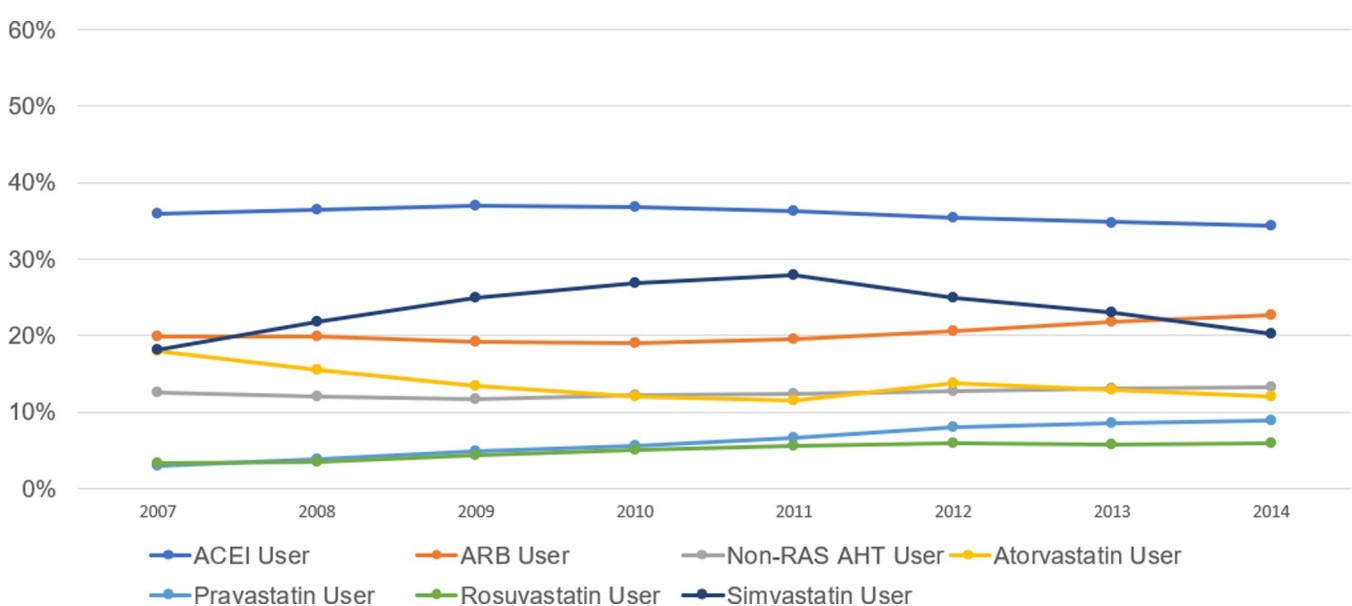

**Fig 2. Percent of Medicare beneficiaries with use of selected statin and antihypertensive prescription drugs, 2007–2014.** Sample is Medicare beneficiaries with fee-for-service and Part D coverage in the year of the horizontal axis. Use of a drug is defined as 90 days and 2 claims. Abbreviations: ACEI (angiotensin converting enzyme inhibitors), ARB (angiotensin-II receptor blockers), RAS (renin-angiotensin system), non-RAS AHTs (beta-blockers, calcium channel blockers, loop diuretics, and thiazide diuretics).

in CYP2C19 confer protection against amyloid-beta burden in persons with AD [54]. Additionally, CYP2C9 is encoded for by a gene previously linked to familial AD [55].

Atorvastatin and simvastatin, while failing to show a protective association against ADRD when combined with ACEIs (compared to other combinations of statins and AHTs), were observed to be protective when combined with ARBs. This might be due to an interaction effect with ARBs on metabolism and transport, similar to what was described for rosuvastatin and ARBs. While they are different in their duration of action (atorvastatin is a long acting statin and simvastatin is a short acting one), both are metabolized by CYP3A4 that has, as with CYP2C9, been shown to have common ARBs as a substrate [53]. The observed variability of benefit may therefore be highlighting underlying differences in some of the drug metabolic processes of these statins. Additionally, these enzymes are encoded by several genes, many of which (e.g., CYP3A4, CYP2D6, CYP2C19) are polymorphic with variable prevalence of different alleles that normally affect metabolizing phenotype in different race/ethnicities [56, 57]; while there is some evidence supporting CYP3A4 variant statin metabolism [56], the clinical significance of these in relation to the drugs studied is unclear. Furthermore, some of the observed differences in protection across race/ethnicity could be caused by differences in the endogenous sodium and renin levels in the peripheral RAS, as was discussed in our earlier work on this subject [44].

The strong protective association of pravastatin is difficult to explain. It is metabolized by glucuronidation rather than cytochrome P450 metabolism [58], and while metabolic changes are often reported in AD [59], it is not clear whether there are any disease-related differences in glucuronidation in AD that might influence how pravastatin is metabolized.

There is some evidence that a net effect of reducing the formation of very low density lipoproteins (VLDLs), precursors of LDLs, can increase levels of LDL receptors [60]. One receptor that might be similarly affected is LRP1, a receptor of VLDLs but also a major factor in receptor-mediated clearance of amyloid-beta in AD [61]. Whether this would be a peripheral or central effect is not determined.

Dyslipidemia and hypertension are both independent risk factors of ADRD, and control of both via lifestyle and medications has been associated with lower incidence and prevalence of ADRD. Recent results from the SPRINT MIND trial found that aggressive blood pressure control (120 mmHg, vs less than 140 mm Hg) was associated with smaller increases in white matter lesion volume [62]. While the present study was not able to address variation in clinical values of blood pressure, we found that hypertension treatment with RAS-acting AHTs plus certain statins may confer ADRD related brain benefits beyond peripheral control of vascular disease.

In addition to controlling blood pressure, RAS medications and to a greater extent ARBs, improve endothelial function, insulin response, and decrease inflammation. Combined statin-based and RAS inhibitor therapies demonstrate additive/synergistic beneficial effects on endothelial dysfunction, insulin resistance, and other metabolic parameters in addition to lowering both cholesterol levels and blood pressure. Mechanistically, the combination of reduced inflammation, improved blood brain barrier integrity, and improved metabolic health all provide a reasonable rationale for reduced risk of ADRD in persons using combined statins and ARBs.

This study has limitations. First, there is a possibility of confounding by indication, wherein individuals who may be differentially predisposed toward ADRD risk are sorted into certain combinations of statins and AHTs. Our study adjusted for many sources of possible bias, but the possibility of unmeasured bias remains. Second, use of AHTs and statins could be imprecisely measured. We analyze robustness of results to different (higher) thresholds of days of use. Drug use in the years prior to Medicare Part D (pre-2006) could confound analyses, if it varies across our comparison groups (different combinations of drugs). Accordingly, we adjust models for years since hypertension and hyperlipidemia diagnoses. Another source of measurement error may be due to individuals switching between molecules or classes within a year, but this would bias results toward zero. Third, timing of ADRD diagnoses in claims data may not represent the true onset of the condition. However, our sample is all users of statins and AHTs and there is no evidence to suggest that diagnostic practices would vary by different combination of statins and AHTs, and therefore imprecise diagnoses would not bias the results. Fourth, the sample is restricted to Medicare beneficiaries enrolled in FFS and Part D (approximately two thirds of Medicare beneficiaries), and thus excludes those in Medicare Advantage or in FFS and not enrolled in Part D, who tend to be slightly healthier on average [45]. Finally, with these data, we cannot test the hypotheses of the various potential mechanisms that may explain the variation by molecule and drug class, and by race and ethnicity that we observed.

## Conclusion

Nearly a quarter of the elderly in the U.S. use both statins and AHTs to treat or prevent dyslipidemia, hypertension, and cardiovascular disease. We found that RAS-acting ARBs combined with pravastatin or rosuvastatin reduced ADRD risk reductions more than statins combined with non-RAS acting AHTs, which may reflect or be dependent on underlying gene polymorphism-associated differences in drug metabolic processes. If these findings are replicated, specific combinations of statins and antihypertensives might be recommended in the interest of reducing ADRD risk. Fewer older adults use pravastatin and rosuvastatin than other simvastatin and atorvastatin. Over time, use of pravastatin and rosuvastatin and ARBs have increased, potentially conferring population benefits of reduced dementia risk.

## Supporting information

**S1 Appendix. Summary of ICD-9 codes used to define various diagnoses and symptoms.**
(DOCX)

**S1 Table. Adjusted odds ratios of AD incidence associated with use of statin-AHT combinations, relative to users of other statin-AHT combinations.** Logistic regression results for AD incidence in sample of 2009–2014 Medicare person-years with 90 possession days and 2 claims of both an AHT and a statin in both years t-1 and t-2. AHTs are antihypertensive (AHT) prescription drugs (angiotensin converting enzyme inhibitors (ACEIs), angiotensin-II receptor blockers (ARBs), beta-blockers, calcium channel blockers, loop diuretics, and thiazide diuretics), and statins are atorvastatin, pravastatin, rosuvastatin, and simvastatin. Sample restricted to person-years with 3 years fee-for-service, 3 years Part D, age 67+, no deaths in the reference year (year t), no prior AD diagnoses, and no prior use of acetylcholinesterase inhibitors (AChEIs) or memantine. Controls are age, age squared, sex, education, income quartiles, statin use (t-1), years since hypertension and hyperlipidemic diagnoses, HCC comorbidity index, number of physician visits, and indicators for past diagnoses of diabetes, atrial fibrillation, acute myocardial infarction, and stroke. Standard errors are clustered at the county level.
(DOCX)

**S2 Table. Adjusted odds ratios of ADRD incidence associated with use of statin-AHT combinations, relative to users of other statin-AHT combinations, with use defined as 180 days and 2 claims.** Logistic regression results for ADRD incidence in sample of 2009–2014 Medicare person-years with 180 possession days and 2 claims of both an AHT and a statin in both years t-1 and t-2. AHTs are antihypertensive (AHT) prescription drugs (angiotensin converting enzyme inhibitors (ACEIs), angiotensin-II receptor blockers (ARBs), beta-blockers, calcium channel blockers, loop diuretics, and thiazide diuretics), and statins are atorvastatin, pravastatin, rosuvastatin, and simvastatin. Sample restricted to person-years with 3 years fee-for-service, 3 years Part D, age 67+, no deaths in the reference year (year t), no prior ADRD diagnoses, and no prior use of acetylcholinesterase inhibitors (AChEIs) or memantine. Controls are age, age squared, sex, education, income quartiles, statin use (t-1), years since hypertension and hyperlipidemic diagnoses, HCC comorbidity index, number of physician visits, and indicators for past diagnoses of diabetes, atrial fibrillation, acute myocardial infarction, and stroke. Standard errors are clustered at the county level.
(DOCX)

**S3 Table. Adjusted odds ratios of ADRD incidence associated with use of statin-AHT combinations, relative to users of other statin-AHT combinations, with use defined as 270 days and 2 claims.** Logistic regression results for ADRD incidence in sample of 2009–2014 Medicare person-years with 270 possession days and 2 claims of both an AHT and a statin in both years t-1 and t-2. AHTs are antihypertensive (AHT) prescription drugs (angiotensin converting enzyme inhibitors (ACEIs), angiotensin-II receptor blockers (ARBs), beta-blockers, calcium channel blockers, loop diuretics, and thiazide diuretics), and statins are atorvastatin, pravastatin, rosuvastatin, and simvastatin. Sample restricted to person-years with 3 years fee-for-service, 3 years Part D, age 67+, no deaths in the reference year (year t), no prior ADRD diagnoses, and no prior use of acetylcholinesterase inhibitors (AChEIs) or memantine. Controls are age, age squared, sex, education, income quartiles, statin use (t-1), years since hypertension and hyperlipidemic diagnoses, HCC comorbidity index, number of physician visits, and indicators for past diagnoses of diabetes, atrial fibrillation, acute myocardial infarction, and stroke. Standard errors are clustered at the county level.
(DOCX)

## Author Contributions

**Conceptualization:** Douglas Barthold, Geoffrey Joyce, Roberta Diaz Brinton, Whitney Wharton, Patrick Gavin Kehoe, Julie Zissimopoulos.

**Data curation:** Douglas Barthold.

**Formal analysis:** Douglas Barthold, Geoffrey Joyce, Julie Zissimopoulos.

**Funding acquisition:** Douglas Barthold, Geoffrey Joyce, Roberta Diaz Brinton, Julie Zissimopoulos.

**Investigation:** Douglas Barthold, Geoffrey Joyce, Roberta Diaz Brinton, Whitney Wharton, Patrick Gavin Kehoe, Julie Zissimopoulos.

**Methodology:** Douglas Barthold, Geoffrey Joyce, Julie Zissimopoulos.

**Supervision:** Geoffrey Joyce, Julie Zissimopoulos.

**Validation:** Douglas Barthold, Geoffrey Joyce, Julie Zissimopoulos.

**Visualization:** Douglas Barthold, Geoffrey Joyce, Julie Zissimopoulos.

**Writing – original draft:** Douglas Barthold, Geoffrey Joyce, Julie Zissimopoulos.

**Writing – review & editing:** Douglas Barthold, Geoffrey Joyce, Roberta Diaz Brinton, Whitney Wharton, Patrick Gavin Kehoe, Julie Zissimopoulos.

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
