## [Decision Letter · Decision Letter 0]

2 Dec 2019

PONE-D-19-25705

Association of combination statin and antihypertensive therapy with reduced Alzheimer’s disease and related dementia risk

PLOS ONE

Dear Barthold,

Thank you for submitting your manuscript to PLOS ONE. After careful consideration, we feel that it has merit but does not fully meet PLOS ONE’s publication criteria as it currently stands. Therefore, we invite you to submit a revised version of the manuscript that addresses the points raised during the review process.

We would appreciate receiving your revised manuscript by Jan 16 2020 11:59PM. To enhance the reproducibility of your results, we recommend that if applicable you deposit your laboratory protocols in protocols.io, where a protocol can be assigned its own identifier (DOI) such that it can be cited independently in the future. For instructions see: http://journals.plos.org/plosone/s/submission-guidelines#loc-laboratory-protocols

We look forward to receiving your revised manuscript.

Kind regards,

Tarek K Rajji

Academic Editor

PLOS ONE

Journal Requirements:

1. Please include captions for your Supporting Information files at the end of your manuscript, and update any in-text citations to match accordingly. Please see our Supporting Information guidelines for more information: http://journals.plos.org/plosone/s/supporting-information.

Reviewers' comments:

Reviewer's Responses to Questions

**Comments to the Author**

1. Is the manuscript technically sound, and do the data support the conclusions?

Reviewer #1: Yes

Reviewer #2: Yes

2. Has the statistical analysis been performed appropriately and rigorously? 

Reviewer #1: Yes

Reviewer #2: Yes

3. Have the authors made all data underlying the findings in their manuscript fully available?

Reviewer #1: Yes

Reviewer #2: Yes

4. Is the manuscript presented in an intelligible fashion and written in standard English?

Reviewer #1: Yes

Reviewer #2: Yes

5. Review Comments to the Author

Reviewer #1: Thank you for the opportunity to review this manuscript. Overall, I was happy with the conduct of the study, the large sample size and robust analysis. The findings have high clinical interest as the problem of Alzheimer’s/related dementias is very prevalent, as is co-morbidity of hypertension and hypercholesterolemia in older people. Given that statins and antihypertensive drugs are widely used, the finding that certain commonly used combinations might offer greater protection against Alzheimer’s/related dementias than others is of great relevance.

There is room for improvement in this manuscript in two areas.

A) Issues in the descriptions relating to the clinical pharmacology of the drugs studied

1) Line 295 – ‘The pharmacology and pharmacokinetics of rosuvastatin are distinct’. First, this is strangely worded as pharmacokinetics are an aspect of pharmacology – this would be better phrased as ‘the pharmacokinetics and pharmacodynamics …’. I would like to see this statement explained further – in what way is rosuvastatin pharmacologically different from the others? Does this extend beyond the pharmacokinetic issues discussed in the following lines to some unique pharmacodynamic/mechanistic property ? I would think that explaining these differences are central to the argument that there is a pharmacological justification underlying the findings.

2) Line 304. The authors talk of the relative potency of the four statins differing from the observed relative benefits in protection against Alzheimer’s – which might mean the protective effect vs Alzheimer’s is mediated by a different mechanism from the LDL-lowering effect. But is potency the key measure here ? Is it not the product of potency and typical dosage – i.e. if drug A is 10 times as potent as drug B, but drug B is typically used at 20 times the dose of drug A, then we would expect drug B, not drug A to offer the most protection (assuming the anti-dementia effect is medicated through the same initial mechanism as the lipid lowering effect)? Presumably also, the statins all have similar efficacy in lipid lowering when used at their standard doses (which therefore offset the differences in potency) or else a literature would have emerged on certain molecules being associated with greater lipid-lowering efficacy than others. So essentially I am not convinced regarding lines the argument from line 304-306.

3) Line 306-313 - The points relating to the CYP metabolizing enzymes are difficult to follow. There appear to be two points which need to be considered separately – i.e. a) there might be CYP-based pharmacokinetic interactions between certain statins and certain antihypertensive agents resulting in higher concentration/dose ratio of one or both drugs which might have an impact on the protective effect against ADRD and b) the impact of statins or antihypertensive drugs on activity of certain CYP enzymes might have a direct impact on risk of ADRD if the presence of the CYP enzyme is itself a risk factor or protective factor for ADRD. Point (a) is weakened by the fact that, according to Flockhart’s table, the statins and antihypertensives are substrates of certain CYP enzymes but are not strong inhibitors so there may often be sufficient metabolizing capacity that the co-prescription of two drugs metabolized by the same CYP may have no impact on their concentrations. As regards point (b) - the genetic variants in CYP expression (such as in CYP2C9 and CYP2C19 which metabolize rosuvastatin) and their ramifications for protection against dementia would already be inherent to the individual whether or not they ever took rosuvastatin (or any other substrate medication), and would only come in to play for the rosuvastatin if perhaps an individual was genetically already a low capacity metabolizer, or co-prescribed a 2C9/2C19 inhibitor, and the presence of rosuvastatin and the appropriate ARB could then inhibit what limited CYP2C9/CYP2C19 activity remained. All in all this is speculative and in my opinion the putative pharmacokinetic mechanisms have been given undue prominence (the CYP suggestions account for over half of the discussion of mechanisms). I would advise organizing this section more logically to cover the (a) and (b) points. The authors may wish to add a small table (or even a bulleted list) derived from Flockhart or similar sources, which allows the user to see which CYPs are involved in the metabolism of the four statins, and the main ARBs and ACE inhibitors and non-ARB antihypertensives, this would allow the potential interactions to be appraised quickly and easily and would demystify this section.

4) Line 324 – I have some reservations with the statement that CYP3A4 is a ‘highly polymorphic enzyme’. It would be reasonable to say ‘polymorphic’. Evidence that this is of functional importance relative to other CYPs is limited. A review by Mizutani T(2003) Drug Metabolism reviews ‘PM Frequencies of Major CYPs in Asians and Caucasians’ pp99-106, stated a Poor Metabolizer frequency for CYP3A4 of 0% in Caucasians (table 2) and 1% in East Asians (table 1), the corresponding figures for CYP2D6 being 7% for in Caucasians and 0.8% in East Asians – while for CYP2C19 19% of East Asians are reported as being poor metabolizers. Looking at the paper the authors referenced (McGraw & Waller, ref 54) it appears that this paper lumps together CYP3A4 with CYP3A5 and describes more heterogeneity with CYP3A5 than CYP3A4. For CYP3A4 it mentions that a 3A4*1B allele is fairly common in African Americans and less so in Caucasians but also states that ‘the clinical impact of this polymorphism has not yet been identified’. There is however some potentially relevant evidence cited in this paper about a CYP3A4*22 allele (presumably associated with an extensive metabolizer phenotype) having impact on simvastatin metabolism. I think the case would be stronger here if you were to also cite and describe the original research relating to this allele (ref 63 Wang , Guo , Wrighton et al who refer to it as a ‘functional SNP located in intron 6’) as this paper provides direct evidence of a CYP3A4-related variant which impacts on statin metabolism.

B) Issues in Sample Description and justification of representativeness

Second, there are some shortcomings in the sample descriptions. In particular – and I think this is of particular relevance to readers not closely familiar with the healthcare system in the USA - the descriptions of the Medicare system (lines 153-158) are inadequate. The general reader needs to know how what is ‘Medicare’ and how is it funded – is this government funded care as is commonly found in Canada and the European Union, is it a wholly privately funded insurance system or some combination of the two? What proportion of US residents/citizens are covered by or registered for Medicare? Was the sample drawn from 20% of Medicare beneficiaries in one specific part of the USA or from 20% of the entire US population registered with correct form of Medicare, aged over 67 and meeting the remaining criteria?

Crucially, there is no discussion of how representative the sample of individuals registered to receive Medicare included in the present study is of the general population in the USA. E.g. Are economically disadvantaged people and ethnic minorities under-represented or over-represented in the population registered to receive Medicare ? The representativeness of the sample must be discussed and if necessary listed as a limitation. I note that in line 271 the sample is described as ‘broadly representative’ but the authors have not given sufficient information to justify this assumption. Further the authors go on (in the discussion) to give further details about the Medicare-based sample using including concepts such as ‘FFS’ (‘fee for service’) and ‘part D’ which is described earlier (line 155) as “part D claims include key elements related to prescription drug events’ - I’m sorry but this is impenetrable to me and will similarly be impenetrable to some readers – these differing gradations of the Medicare system need to be explained in more generic terms for readers unfamiliar with your system. What (line 366) is the ramification for the respresentativeness of the sample of excluding people in FFS without part D or in Medicare advantage? These concepts must be defined in the methods section or it is impossible for the reader to judge.

C) Further Question

Do the authors consider that there is a possibility of ‘confounding by indication’ – i.e. certain patient characteristics encouraging prescribers to prefer or avoid certain drugs in patients with attributes which might themselves be risk factors for dementia. One example might be using beta blockers preferentially as an antihypertensive drug of choice in patients exhibiting longstanding anxiety which could, through its associations both with depression and cardiovascular disease be associated with development of dementia. I am not suggesting that this is a sufficiently important issue that it demands that some mitigating methodology such as propensity-score matching should have been used, but I think the possibility of this occurring should be mentioned.

D) Minor points

1) In the Abstract – ‘FINDINGS’ section, the phrase “Pravastatin and rosuvastatin combined… “ is ambiguous, would be better phrased as “Pravastatin OR rosuvastatin combined…”

2) In the Abstract – ‘METHODS’ section – the last sentence is a little confusing and would be improved by changing the final ‘and’ to “vs.”

3) In the Abstract – ‘CONCLUSION’ section – I suggest changing ‘… is less common than other statins’ to is ‘is less common than use of simvastatin or atorvastatin. This is because other statins exist which are not mentioned in this manuscript, e.g. fluvastatin and lovastatin. This also applies in line 374 (conclusion).

4) In the Introduction, lines 121 – 123. I would suggest providing more information on the observational studies rather than merely referencing that they exist. Being picky, it is actually incorrect to say that they are all ‘more recent’ than the RCTs. But I believe the reader would be better served if there could be some greater detail and critique about these studies (at least those that the authors view as the most important), as understanding what they have found is essential to the context of the present study.

5) Line 135 – in defining ‘other AHT drug classes’ only calcium channel blockers, beta blockers and thiazides are mentioned – note the bracket is an ‘i.e.’ (suggesting there should be a complete list) and not an ‘e.g.’. The obvious missed class is loop diuretics which as I understand are still commonly used and were included in this study. What also of other less common AHT classes – alpha blockers, mixed alpha/beta blockers (e.g. carvidelol), vasodilators, centrally acting alpha-2 agonists? Can the authors add a line (around line 175) in the methods to justify excluding these less commonly used drug classes?

6) Line 293 – can the authors clarify if the loss of patent protection for Crestor/Rosuvastatin described in 2016 was specific to the United States?

7) Line 320 – I wonder if ‘CYP2CP’ is a typo – the referenced Flockhart table (ref 53) lists CYP2C9 and CYP2C19 as well as the less familiar CYP2C8, presumably this should be CYP2C9? Alternatively perhaps this is not a typo and the authors are using ‘CYP2CP’ as an umbrella term to refer to all the CYP2C enzymes - this would however be an unusual usage. Either way this needs clarification.

8) Conclusion – I would suggest adding a sentence to state that these findings, if replicated, suggest a possibility that certain antihypertensive-statin combinations might be recommended preferentially to older adults in the interest of reducing risk of ADRD.

Reviewer #2: This is an exciting paper using large databases of Medicare beneficiaries to examine the combination of statin and antihypertensive therapy (AHT) and related dementia risk. This is an important topic, because a common recommendation for patients to prevent dementia is to take statins and AHTs, although real-world data with large enough sample size to assess the effects of individual statin/AHT combos is limited. I only have some major and minor comments:

1. Is the overall combo of statin (any) + AHT (any) has a beneficial association with dementia, relative to a non-AHT, non-statin comparator? This could be reported in the abstract - since a clinician reading this paper, may get a "blanket message" that any combo is good - whereas perhaps only certain combinations are helpful

2. It's not clear in the abstract what your primary question is - whether (or not) any combo of statin(any)/AHT(any) will be associated with improved outcome (this is what seems to be); and if not - whether you are particularly interested in certain drug combinations (and which of those is your primary question)

3. Even though this is a large set - it's not clear whether the sample size was necessarily large enough to test each of the combinations separately - If this is the case, it should be reported in the limitations section.

4. Study Design - This is not clear - is this a cohort study (which it seems to be) or is it a nested case control design or other design? Also it is not clear how long the maximum follow up (seems to be 5 years, but can be made more explicit)? Did you only include people with 2 years of AHT/statin use? What happened to people with shorter durations of use - were they excluded completely (ideal) or put in the reference group?

5. Another important limitation of the study is that it used a prevalent user (of AHT/statin) cohort - as opposed to a new-user design (previously unexposed to AHT/statin), which weakens causality. This should be mentioned in the limitations

Minor comment:

6. Sometimes ORs are reported in three digits - OR=0.942 - the industry standard is sometimes 0.94 in such cases. The journal could let you know if they have a preference for how to report such statistcis.

7. Ideally a statistician reviewer from the journal should also double-check this paper.

6. PLOS authors have the option to publish the peer review history of their article (what does this mean?). If published, this will include your full peer review and any attached files.

Reviewer #1: No

Reviewer #2: No

---

## [Author Response · Author response to Decision Letter 0]

16 Jan 2020

Our responses to reviewers are in the response to reviewers file.

---

## [Decision Letter · Decision Letter 1]

10 Feb 2020

Association of combination statin and antihypertensive therapy with reduced Alzheimer’s disease and related dementia risk

PONE-D-19-25705R1

Dear Dr. Barthold,

We are pleased to inform you that your manuscript has been judged scientifically suitable for publication and will be formally accepted for publication once it complies with all outstanding technical requirements.

With kind regards,

Tarek K Rajji

Academic Editor

PLOS ONE

Additional Editor Comments (optional):

Reviewers' comments:

Reviewer's Responses to Questions

**Comments to the Author**

1. If the authors have adequately addressed your comments raised in a previous round of review and you feel that this manuscript is now acceptable for publication, you may indicate that here to bypass the “Comments to the Author” section, enter your conflict of interest statement in the “Confidential to Editor” section, and submit your "Accept" recommendation.

Reviewer #1: All comments have been addressed

Reviewer #2: All comments have been addressed

2. Is the manuscript technically sound, and do the data support the conclusions?

Reviewer #1: Yes

Reviewer #2: Yes

3. Has the statistical analysis been performed appropriately and rigorously? 

Reviewer #1: Yes

Reviewer #2: I Don't Know

4. Have the authors made all data underlying the findings in their manuscript fully available?

Reviewer #1: Yes

Reviewer #2: Yes

5. Is the manuscript presented in an intelligible fashion and written in standard English?

Reviewer #1: Yes

Reviewer #2: Yes

6. Review Comments to the Author

Reviewer #1: Thank you. I have gone through the authors’ changes and am happy with them – relating in particular to the descriptions of clinical pharmacology of the drugs studied and to the sample description/justification of representativeness (including the much improved descriptions of the Medicare-based enrollment). I have no further major comments.

Minor points

1) Please note there is a minor typo in line 315: ‘t’ should be ‘to’.

2) Line 388-389 Re: ‘Our study adjusted for many sources of possible bias that may lead to differential use of certain combinations of statins and AHTs, but the possibility of unmeasured bias remains.’ – I would invite the authors to consider whether strictly speaking this statement is referring to ‘confounding’ rather than ‘bias’.

Reviewer #2: The authors appear to have appropriately responded to my questions regarding the manuscript. I have no further recommendations.

7. PLOS authors have the option to publish the peer review history of their article (what does this mean?). If published, this will include your full peer review and any attached files.

Reviewer #1: Yes: Simon J C Davies

Reviewer #2: No

---

## [Editor Report · Acceptance letter]

18 Feb 2020

PONE-D-19-25705R1 

Association of combination statin and antihypertensive therapy with reduced Alzheimer’s disease and related dementia risk 

Dear Dr. Barthold:

I am pleased to inform you that your manuscript has been deemed suitable for publication in PLOS ONE. Congratulations! Your manuscript is now with our production department. 

With kind regards,

on behalf of

Dr. Tarek K Rajji 

Academic Editor

PLOS ONE